# The Cluster Description Problem - Complexity Results, Formulations and Approximations

**Ian Davidson**[*]
Department of Computer Science
University of California - Davis
`davidson@cs.ucdavis.edu`

**Antoine Gourru**
Universite de Lyon (ERIC, Lyon 2)
`antoine.gourru@univ-lyon2.fr`

**S. S. Ravi**[†]
Biocomplexity Institute
University of Virginia
`ssravi0@gmail.com`

## Abstract

Consider the situation where you are given an existing $k$-way clustering $\pi$. A challenge for explainable AI is to find a compact and distinct explanation of each cluster which in this paper is assumed to use instance-level descriptors/tags from a common dictionary. Since the descriptors/tags were not given to the clustering method, this is not a semi-supervised learning situation. We show that the *feasibility* problem of testing whether any distinct description (not necessarily the most compact) exists is generally intractable for just two clusters. This means that unless **P = NP**, there cannot exist an efficient algorithm for the cluster description problem. Hence, we explore ILP formulations for smaller problems and a relaxed but restricted setting that leads to a polynomial time algorithm for larger problems. We explore several extensions to the basic setting such as the ability to ignore some instances and composition constraints on the descriptions of the clusters. We show our formulation's usefulness on Twitter data where the communities were found using social connectivity (i.e. `follower` relation) but the explanation of the communities is based on behavioral properties of the nodes (i.e. hashtag usage) not available to the clustering method.

## 1 Introduction and Motivation

There are many clustering algorithms which perform well towards their aim of finding cohesive groups of instances. The Louvain method [2] consistently generates useful results for graph clustering, spatial data clustering methods such as DBScan [8] are used extensively for geographical data problems and a plethora of clustering methods targeted towards images, documents and graphs exist [12, 22, 5, 20, 6]. However, a growing need for machine learning methods is the need for explainability. Here, we explore the idea of taking an *existing* clustering defined by a partition $\pi = \{C_1, C_2, \ldots, C_k\}$ found using data set $X$ and explaining it using another data set $Y$. For example, $X$ could be (as they are in our experiments) the $n \times n$ adjacency matrix of a graph showing the *structural* relation between individuals (e.g. the `follower` relation in Twitter) and $Y$ an $n \times t$ *behavioral* information matrix showing how often each individual posted on each of $t$ different hashtags. Importantly only $X$ and *not* $Y$ was used to find the clustering; hence, this is not a semi-supervised setting. This situation

---

[*]Institute of Advanced Studies Fellow 2017-2018 at Collegium de Lyon. Supplementary material and source code available at `www.cs.ucdavis.edu/~davidson/description-clustering`.

[†]Also with Dept. of Computer Science, University at Albany – State University of New York.

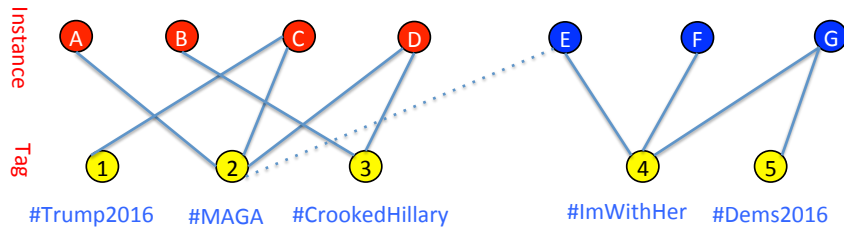

Figure 1: A simple twitter network example where there are two (red/blue) clusters/communities to be explained using the hashtags each individual uses. For example person A uses `MAGA`, person B `CrookedHillary` and so on. In this simple example (without the dotted line) the RED cluster is compactly covered/explained by tags {`MAGA, CrookedHillary`} and the BLUE cluster with {`ImWithHer`}. But instance $E$ only having tag `MAGA` (indicated by the dotted line) produces a more complex example which there is no feasible solution. One approach we explore is to ignore/omit instances such as $E$ which yields an explanation RED {`MAGA, CrookedHillary`} and BLUE {`ImWitHer`}.

where the clustering is completed on one view of the data (e.g. detailed satellite imagery) and needs to be explained using another (e.g. high level land usage information) is common in many fields.

Throughout this paper we describe our work using a twitter example, but envision it could be used in many different settings of which it is an example of the first. *Setting #1:* Clusters are found using complex features which are good for clustering but poor for explanation; hence, explanation is via easy to understand tags. For images, clusters found using SIFT features can be explained via tags based on caption information or user specified keywords. For social networks, clusters found on graph topology can be explained using node attributes (e.g. hash tag usage, demographics). Finally, clusters found on electronic healthcare records can be explained using symptoms, diseases or healthcare process events [21]. *Setting #2:* Clusters were historically created by humans (or nature) but a modern explanation is required. For instance, consider electoral maps where electoral districts were grouped into constituencies for historic or geographic reasons, perhaps decades ago [15]. We may now wish to explain the clustering using current demographic information of the population. *Setting #3: Clusters are formed using private/sensitive data but explained using public data.* This setting is common in security problems so as to not compromise privacy [14] by describing clusters using the centroids which are based on private data. This is a variant of Setting #1 as the private data is good for clustering but inappropriate for explanation.

Consider the simple example shown in Figure 1 where the twitter accounts are clustered (say based on the follower relationship) into two clusters and are to be explained using their hashtag usage. We want one group of hashtags to cover all the instances in the blue cluster and another set for the red cluster; however, to differentiate and explain the clusters there should be no or minimal overlap between the two explanations. It is clear there is a set cover aspect to the problem, however, regular set cover is insufficient for several reasons: (i) there are multiple universes (one for each cluster) to cover, (ii) each universe should be covered by non-overlapping (or minimally overlapping) collection of tags and (iii) there may be composition constraints on the tags to be used. For example, we may have background information of the form: (a) if the hashtag `#MAGA` is used to describe a cluster then `#MakeAmericanGreatAgain` should also be used or (b) `#MAGA` cannot be used together in a descriptor that has `#ImWithHer`. Such background information could be generated from domain experts or automatically from say hashtag co-usage statistics.

We formulate the cluster description problem in Section 3 and show that it is intractable even for two clusters. We formulate an ILP version of the problem in Section 4 and observe that the number of variables and constraints is linear with respect to the number of instances, tags and clusters. However, in practice this formulation was found to be restrictive and sometimes the explanations contain inconsistencies or were incomplete. We therefore explore two extensions: (i) a formulation where we can ignore a given number of the instances to ensure a feasible solution and (ii) a formulation where we can add in background information. Our experiments show these formulations can handle clusterings of tens of thousands of instances explained using hundreds of tags on laptops. For larger

problems in Section 5 we explore an $\alpha - \beta$ relaxation for which a polynomial time algorithm exists where each cluster's description has at most $\alpha$ tags and no two cluster descriptions can have more than $\beta$ tags in common. Section 6 shows a variety of different uses of our method on Twitter networks. We conclude in Section 7.

The contributions of the paper are as follows.

(1) We formulate the novel cluster description problem as a combinatorial optimization problem to find concise but different descriptions of each cluster.

(2) We show that the feasibility problem of just finding any cluster description is intractable even for two clusters (Theorem 3.1).

(3) Our basic ILP formulation models complex forms of set cover involving multiple universes (one for each cluster) and extended formulations explore ignoring instances (which are in some sense outliers) and composition constraints on the tags. To our knowledge these are novel.

(4) We construct a sufficient condition when the cluster description problem is in **P** (see Theorem 5.1) and construct an algorithm for that setting (see Algorithm 1). We term this $\alpha - \beta$-CONS-DESC since each description must be at most length $\alpha$ and no two descriptions can overlap by more than $\beta$ tags.

## 2    Related Work

There is a considerable body of literature on *simultaneously* performing clustering and finding an explanation. Work such as conceptual clustering [11, 9] attempts to use the **same features** to find the clustering and explain it and hence is limited to a single view of categorical variables. More recent work has explored conceptual clustering extensions using constraint programming [13], SAT [17] or ILP [18, 19] where the objective is to find a clustering and explanation simultaneously. Predictive clustering [16] uses the features to find a clustering and a descriptive function for the purpose of prediction and is not unlike a decision tree. All of this work is different from our work since: (i) the goal is to create a new clustering not explain an existing one and (ii) the explanation uses features which were used to perform the clustering. Multi-view version of conceptual-clustering style learning has been studied (e.g. [4]); these algorithms also attempt to find a clustering but not to explain it and do not scale due to the Pareto optimization formulation.

## 3    The Cluster Description Problem

We are given a set $S = \{s_1, s_2, \ldots, s_n\}$ of $n$ items and a partition $\pi$ of $S$ into $k$ clusters $C_1, C_2, \ldots,$ $C_k$. We are also given a universal set $T$ of tags, and for each item $s_i$, a set $t_i \subseteq T$ of tags, $1 \leq i \leq n$. The goal is to find a subset $T_j \subseteq T$ of tags for each cluster $C_j$ ($1 \leq j \leq k$) such that all the following conditions are satisfied.

(a) For each cluster $C_j$ and each item $s_i \in C_j$, $T_j$ has at least one of the tags in $t_i$; formally, $|T_j \cap t_i| \geq 1$, for each $s_i \in C_j$ and $1 \leq j \leq k$.

(b) The sets $T_1, T_2, \ldots, T_k$ are pairwise disjoint.

For $1 \leq j \leq k$, the set $T_j$ will be referred to as the **descriptor** for cluster $C_j$. We will refer to the above problem as the **Disjoint Tag Descriptor Feasibility** (DTDF) problem. In this version, no constraints are imposed on the number of tags in any descriptor; any collection of descriptors satisfying conditions (a) and (b) above is acceptable. Later in Section 4 we will cover the **Disjoint Tag Descriptor Minimization** (DTDM) problem which adds the requirement that the size of the description is minimized, that is:

(c) $\sum_j |T_j|$ is minimized.

### 3.1    Complexity of DTDF

This section and the related part of the supplementary material can be skipped on a first reading of the paper with the understanding that the following theorem implies the computational intractability of DTDF and hence DTDM; that is, no polynomial time algorithms can exist for them, unless **P**

**= NP**. All versions of DTDF can be reduced to SAT and this allows us to identify some restricted versions of DTDF that can be solved efficiently (see Section 7).

**Theorem 3.1** *The* DTDF *problem is **NP**-complete even when the number of clusters is 2 and the tag set of each item is of size at most 3.*

**Proof:** See supplementary material.

## 4 An ILP Formulation for the DTDM Problem

We first sketch our basic formulation for the DTDM problem and then introduce enhancements.

**Basic Formulation.** We are given a clustering $C_1, C_2 \ldots C_k$ of $n$ instances with each instance described by a subset of the $t = |T|$ tags. These tags are in the $n \times t$ matrix $Y$. We solve for the $k \times t$ binary matrix $X$ where $X_{i,j} = 1$ iff cluster $i$ is described by tag $j$. One objective function then is simply find the most concise overall cluster description:

$$argmin_X \sum_{i,j} X_{i,j} \tag{1}$$

Hence the number of variables in this version of the formulation is $kt$ where $k$ is the number of clusters and $t$ is the number of tags.

Our first basic constraint includes the set coverage requirement for each different cluster/universe. Here we must define the matrices $S^1, \ldots, S^k$, where $S^a_{i,j} = 1$ iff the $i^{th}$ instance is actually in cluster $a$ and has tag $j$. Note that $S^i$, $1 \le i \le k$, can be pre-computed. Since each instance must be explained/covered there will be $n$ constraints of this type.

$$s.t. \ \sum_j X_{k,j} S^k_{i,j} \ \ge \ 1 \ \forall \, i \in C_k, \, \forall \, k \tag{2}$$

Our next basic constraint requires that the tags chosen to represent each cluster do not overlap that is they must be disjoint ($w_j = 1$) or minimally overlap ($w_j > 1$), where $w_j$ is the maximum number of times tag $j$ can be used in descriptors. This is simply an `OR` constraint and can be encoded as:

$$s.t. \ \sum_i X_{i,j} \ \le \ w_j \ \forall \, j \tag{3}$$

There will be $t$ constraints of this type where $t$ is the number of tags. So overall the number of variables to solve for is $O(tk)$ and the total number of constraints is $O(n + t)$.

**Extended Formulation.** The previous formulation meets the requirements of finding a concise and different description of each cluster. However, in practice we found several limitations. Firstly, as the intractability result shows, finding just a feasible solution is challenging and often in experiments the solver did not converge to a feasible solution. Making $w_j$ larger could address this problem but then the descriptions of the clusters become more similar to each other reducing their usefulness. Secondly, when the solver did return a solution, the descriptors returned for each cluster were sometimes *incomplete* or *inconsistent*. In the former category a cluster could be described by #MAGA but not #MakeAmericaGreatAgain and in the later category a cluster could be described by both #MAGA and #IamWithHer.

To address these concerns we explore three additions. The first two allow side-stepping the infeasibility issue by relaxing the strict requirements of the description. The first such addition allows one tag to describe multiple clusters and the second allows ignoring some instances. The third addition incorporates composition constraints to ensure that the descriptions match human or machine-generated guidance.

**Minimizing Overlap.** Rather than minimizing the total description length we can allow the same descriptor/tag to describe multiple tags but attempt to minimize these overlaps. This can be achieved by having the objective:

$$argmin_C \sum_j w_j \tag{4}$$

The number of variables in this version of the formulation is still $kt$. It is possible to combine this objective with the objective defined in Equation (1).

**Cover-or-Forget.** To our basic coverage requirement (Equation (2)) we add in the ability to *forget* $I_i$ instances for cluster $i$. To model this, we introduce the set of variables $Z$ where $z_i = 1$ iff instance $i$ is *ignored*. This can be encoded by replacing Equation (2) above with Equations (5) below. Note this introduces $n$ more optimization variables, bringing the total number to $O(tk + n)$.

$$s.t. \ \ z_i + \sum_j X_{k,j} S_{i,j}^k \geq 1 \ \ \forall \, i \in C_k, \ \forall \, k \tag{5}$$

$$s.t. \ \ \sum_i z_i \ \leq \ I_k \ \ \forall i \in C_k, \ \forall \, k$$

**Composition Constraints.** To require two tags to always be used to describe the same cluster or two tags to never describe the same cluster we introduce two sets of pairs, namely `Together` and `Apart`, which are not unlike the must-link and cannot-link constraints used in constrained clustering [7, 1] though the complexity results are different (see Section 7). This adds the further constraints:

$$s.t. \ \ X_{k,i} + X_{k,j} \leq 1 \ \ \forall \, \{i,j\} \in \texttt{Apart}, \ \forall \, k \tag{6}$$
$$s.t. \ \ X_{k,i} = 1 \rightarrow X_{k,j} = 1 \ \ \forall \{i,j\} \in \texttt{Together}, \ \forall \, k \tag{7}$$

The latter constraint is non-linear but can easily be modeled by merging two hashtags into one which simply involves merging columns in $C$ and $S$.

## 5 A Relaxed Setting and Polynomial Time Algorithm

**A Note on Terminology.** Throughout this section, we say that certain parameters of a problem are "fixed" and others are "not fixed". These notions are commonly used in complexity theory [10]. The reader familiar with these concepts can skip this discussion but we provide a brief review here since the ideas are crucial in understanding our results. Consider the Minimum Set Cover (MSC) problem [10]: given a universe $U = \{u_1, u_2, \ldots, u_n\}$ with $n$ elements, a collection $W = \{W_1, W_2, \ldots, W_m\}$ with $m$ subsets of $U$ and an integer $k \leq m$, is there is a subcollection $W'$ of $W$ such that $|W'| \leq k$ and the union of the sets in $W'$ is equal to $U$? In the version of this problem where $k$ is *not* fixed (and which is **NP**-complete [10]), the value of $k$ is not known a priori, so no pre-computations are possible. In the fixed parameter version (which is in **P**), $k$ is a known constant (such as 4); so, one may use pre-computation.

### 5.1 Results for the $(\alpha, \beta)$-CONS-DESC Problem

As shown in Section 3, one of the reasons for the computational intractability of the DTDF problem is the requirement that the cluster descriptors be pairwise disjoint. We now consider a version of the descriptor problem, which we call $(\alpha, \beta)$-CONS-DESC (for "$(\alpha, \beta)$-constrained descriptor" problem), where the disjointness requirement is relaxed. Here, in addition to the clusters and the tag sets for each instance, we have two *fixed* integer parameters $\alpha$ and $\beta$, and the requirements are as follows: (i) for each cluster $C_j$, the descriptor for $C_j$ must have a nonempty intersection with the tag set of each instance in $C_j$, (ii) each descriptor must have at most $\alpha$ tags and (iii) no two descriptors may have more than $\beta$ tags in common.

We first show that the $(\alpha, \beta)$-CONS-DESC problem can be solved in polynomial time when, in addition to $\alpha$ and $\beta$, the number $k$ of clusters is *also fixed*. We next show that the condition on $k$ cannot be relaxed; that is, when $k$ is not fixed, the $(\alpha, \beta)$-CONS-DESC problem is **NP**-complete even when $\alpha = 4$ and $\beta = 1$.

**Theorem 5.1** *The $(\alpha, \beta)$-CONS-DESC problem can be solved in polynomial time when the number of clusters $k$ is fixed. This algorithm can also handle* `Together` *and* `Apart` *composition constraints.*

**Proof:** The idea is to enumerate all the possible descriptors for each cluster systematically; the fixed values of $\alpha$ and $k$ ensure that the running time of the algorithm is a polynomial function of the input size. (The steps of the described in the proof are shown in Algorithm 1.)

Let $N$ denote the maximum number of tags used in any cluster. (Note that $N \leq |T|$, where $T$ is the universal set of all descriptors.) Since the descriptor for a cluster must have at most $\alpha$ tags, the number of possible descriptors for each cluster is $\binom{N}{\alpha} = O(N^\alpha)$. Call a cluster descriptor **valid** if it satisfies all the given `Together` and `Apart` composition constraints. Since the size of each descriptor is at most $\alpha$, the number of constraints is $O(\alpha^2)$. Thus, checking whether all the given constraints are satisfied can be done in $O(\alpha^2)$ time. Since $\alpha$ is fixed, $O(\alpha^2) = O(1)$. Suppose we choose one valid descriptor for each cluster. Let $(D_1, D_2, \ldots, D_k)$ be a $k$-tuple of descriptors, where $D_j$ is the chosen valid descriptor for cluster $C_j$, $1 \leq j \leq k$. Since we need to consider only $O(N^\alpha)$ descriptors for each cluster and the number of clusters is $k$, the number of $k$-tuples of descriptors to be considered is $O([N^\alpha]^k) = O(N^{k\alpha})$. For each such $k$-tuple, we can efficiently check whether there is any pair of descriptors that share more than $\beta$ tags. If there is no such a pair, we have a solution and the algorithm terminates; otherwise, we discard the current $k$-tuple and consider the next $k$-tuple of descriptors. If none of the $k$-tuples is a solution, the algorithm terminates after indicating that there is no solution.

The steps are shown in Algorithm 1 where we have assumed that the existence of an algorithm that maintains a circular list of valid descriptors (each of size at most $\alpha$) for each cluster and returns the next valid descriptor from the list whenever the statement "Get the next descriptor" is executed.

---

**Algorithm 1:** Description of our Algorithm for $(\alpha, \beta)$-CONS-DESC

---

**Input** : A collection of $k$ clusters $C_1, C_2, \ldots, C_k$ with tag sets for each instance in each cluster.
**Output** : A valid descriptor with at most $\alpha$ tags for each cluster such that any pair of descriptors have at most $\beta$ tags in common. (Please see the main text for the definition of a valid descriptor.)

1  **for** Cluster $C_1$ **do**
2      Get the next valid descriptor $D_1$.
3      **for** Cluster $C_2$ **do**
4          Get the next valid descriptor $D_2$.
5                  $\vdots$
6          **for** Cluster $C_k$ **do**
7              Get the next valid descriptor $D_k$.
8              Let $\mathcal{D} = (D_1, D_2, \ldots, D_k)$.
9              **if** *Each pair of descriptors in $\mathcal{D}$ have at most $\beta$ tags in common* **then**
10                 Output $\mathcal{D}$ as the solution and **stop**.
11             **end**
12         **end**
13     **end**
14 **end**
15 Print "No solution".

---

The correctness is obvious since the algorithm tries all possible combinations of valid descriptors. To estimate the running time, note that the algorithm considers $O(N^{k\alpha})$ $k$-tuples of descriptors. For each $k$-tuple, it considers $\binom{k}{2} = O(k^2)$ pairs of descriptors. For each pair of descriptors, a simple search that uses $O(\alpha^2)$ time is sufficient to determine whether the pair has more than $\beta$ tags in common. Since $\alpha$ is fixed, $O(\alpha^2) = O(1)$. Thus, the overall running time is $O(N^{k\alpha} k^2)$, which is polynomial since $k$ and $\alpha$ are fixed. ∎

Our next result shows that when the number of clusters $k$ is *not fixed*, the $(\alpha, \beta)$-CONS-DESC problem remains **NP**-complete. Our proof of the following result appears in the supplement.

**Theorem 5.2** *When the number of clusters $k$ is not fixed, The $(\alpha, \beta)$-CONS-DESC problem is **NP**-complete even when $\alpha = 4$ and $\beta = 1$.* ∎

## 6 Experimental Results with Twitter Election Data

A series of easy to use MATLAB functions encoding our three formulations is available at `www.cs.ucdavis.edu/~davidson/description-clustering`.

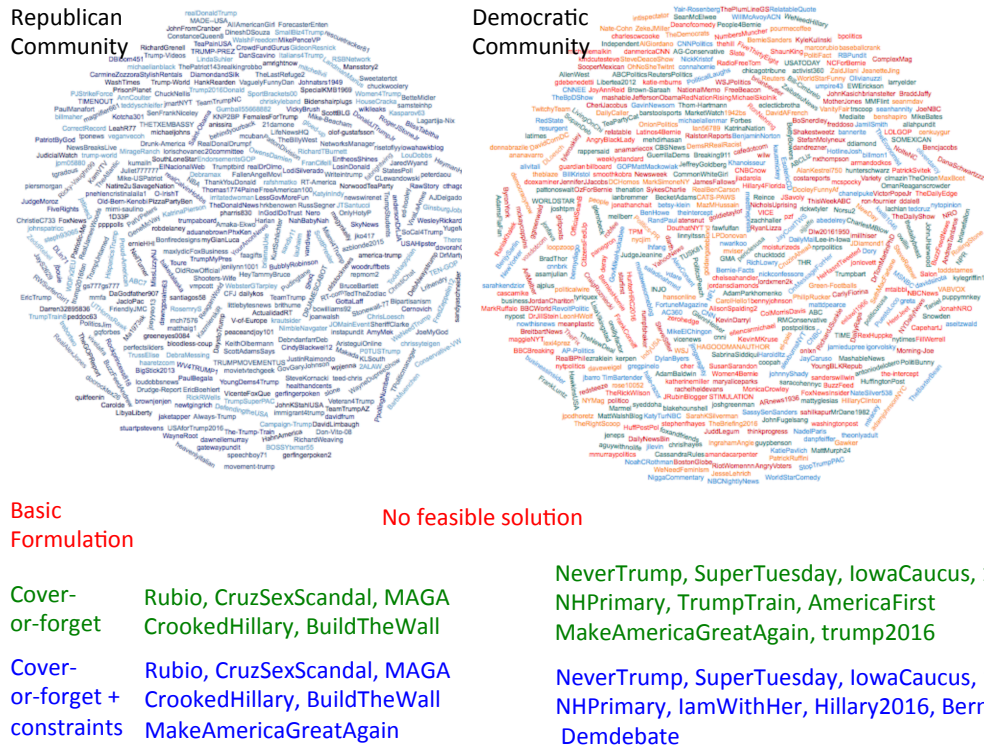

Republican Community

Democratic Community

Basic Formulation

No feasible solution

Cover-or-forget: Rubio, CruzSexScandal, MAGA CrookedHillary, BuildTheWall

NeverTrump, SuperTuesday, IowaCaucus, 1, NHPrimary, TrumpTrain, AmericaFirst MakeAmericaGreatAgain, trump2016

Cover-or-forget + constraints: Rubio, CruzSexScandal, MAGA CrookedHillary, BuildTheWall MakeAmericaGreatAgain

NeverTrump, SuperTuesday, IowaCaucus, 1, NHPrimary, IamWithHer, Hillary2016, Bernie Demdebate

Figure 2: A Twitter Network of the 1000 most popular accounts divided into two communities using spectral clustering explained by their use of political hashtags during the 2016 US primary election season. The basic formulation was too restrictive and no feasible solution exists. The cover-or-forget formulation finds a solution for $I_1 = I_2 = 5$ where no users were ignored in the Republican community but the following users were ignored in the Democratic community: `ZaidJilani`, `VictorPopeJr`, `TedTheZodiac`. The cover-or-forget + constrained formulation finds more complete results for the Republican community and more consistent results for the Democratic community.

**Illustrative Results With Spectral Clustering.** We use an experimental data set of Twitter that we collected. The Twitter data was collected from 01/01/16 until 08/22/16 and covers the political primary season of the United States 2016 Presidential Election. The 1000 most politically active twitter users were chosen and a graph $X$ was constructed based on their retweet behavior. That is, $X_{i,j}$ (the weight of the edge $\{i, j\}$) is the number of times node $i$ is retweeted by node $j$. Also, the 136 most used political hashtags were collected to obtain $Y$.

In this first experiment we use spectral clustering to divide $X$ into just two communities and we found two natural (and obvious) communities amongst follower information: pro-Democratic and pro-Republican. Attempting to find two distinct explanations (with no overlap) from the 136 hash tags yields no feasible solution using our basic formulation. Instead we used our cover-or-forget formulation setting $I_1 = I_2 = 5$ so that some instances could be ignored. However, this produced inconsistent results when covering the pro-Democratic community as `MakeAmericaGreatAgain` and `NeverTrump` are used to cover the same community! We calculated the frequency of co-occurrence of the hashtags in the same tweet, and from the top 2% generated `Together` constraints and from the bottom 2% generated `Apart` constraints. Though this method is crude, the results are promising but we are sure a human could do a better job at generating these constraints. Results are shown in Figure 2. Interestingly the Republican community has a simpler explanation whilst the Democratic community's explanation was longer and less focused.

**Experiments with the Louvain Method.** Here we take a data set as before except that we expand it to have ≈ 5000 of the most popular twitter accounts who were politically active. The Louvain

| Community | Description |
|---|---|
| Pro-Clinton | `NeverTrump ImWithHer DemDebate Sanders p2` |
| Pro-Sanders | `DemsInPhilly IowaCaucus FeelTheBern DonaldTrump` |
| Pro-Trump | `Trump SuperTuesday MakeAmericaGreatAgain` |
| Pro-Cruz | `GOPDebate Cruz Clinton Breaking` |
| Other | `GOP,BernieSanders` |

Table 1: The four main communities found by the Louvain method (the fifth is an amalgamation of the smaller communities) on the Retweet graph and their description using hashtags.

| Community | Description |
|---|---|
| January-February | `GOPDebate, Trump2016, Cruz, tcot, VoteTrump2016, Iowa, NH` |
| March-April | `LyinTed, CruzSexScandal, BuildTheWall, Alaska, Arkansas, Oklahoma, Texas` |
| May-June | `BuildTheWall, Indiana, Washington` |
| July-August | `MakeAmericaGreat, MAGA, CrookedHillary, Benghazi` |

Table 2: The pro-Trump communities behavior explained for four pairs of months of the primary season: Jan 2016 to August 2016.

method divides the `retweet network` (where the edge weight is simply how often node $i$ retweets a message from node $j$). The Louvain method discovers many clusters but the 4 largest naturally correspond to: `pro-Clinton`, `pro-Sanders`, `pro-Trump`, `pro-Cruz` and `Other` which is a combination of many small communities. We attempt to describe these communities using the 136 most popular political tags and allow each tag to only appear once in each cluster. The results are in Table 1. For the Pro-Trump, Pro-Clinton, Pro-Cruz and Pro-Sanders communities, the results are as expected referring to their candidate, the opposition and slogans.

**Experiments with Evolving Behavior.** Here we take the previously found Pro-Trump community in Figure 1 and create four versions of its behavior from its Hashtag usage in January/February, ..., July/August. Applying our method to this setting allows us to explain the different/evolving behavior of one community over time. Results are shown in Table 2.

**Experiments on Scalability.** Here we explore the run time of our solver on a modest computing platform (single core of a 2016 MacBook Air) using the MATLAB solver (`intlinprog`). No doubt faster computation times could be obtained by using state of the art clusters and solvers such as Gurobi and CPLEX but we wish to explore trends on the computation time as a function of the size of the number of instances and number of tags. Table 3 show results for varying number of instances (left) and number of tags (right). It is important to realize that as the number of nodes in the graph becomes larger but the number of tags is a constant, we need to forget more nodes to find a feasible explanation. From these experiments we can conclude the ILP formulations are useful for problems of tens of thousands of instances and hundreds of tags. For larger problems, approximation formulation of Section 5 would need to be used.

| # Nodes | k | Time (s) | Nodes Forgotten | # Tags | k | Time (s) | Nodes Forgotten |
|---|---|---|---|---|---|---|---|
| 1000 | 2 | 0.5 | 5 | 25 | 32 | 232.1 | 783 |
| 2000 | 4 | 1.1 | 46 | 50 | 32 | 153.5 | 325 |
| 3000 | 8 | 15.3 | 56 | 75 | 32 | 143.1 | 178 |
| 4000 | 16 | 32.6 | 89 | 100 | 32 | 123.4 | 155 |
| 5000 | 32 | 88.1 | 123 | 136 | 32 | 88.1 | 123 |

Figure 3: Runtime of basic intlinprog matlab solver on a single core of a 2016 MacBrook Air. Left table shows the solver time as a function of graph size and the right table as a function of tag size. For the right table the number of tags is the most frequent tags to increase the likelihood of finding a solution.

# 7 Additional Results and Conclusion

Here we mention some interesting but not critical results we did not have space for in the paper, sketch future work and conclude. There are simple sufficient conditions when the DTDF and DTDM problems can be solved efficiently. The conditions arise from the fact that for two clusters, the problems can be easily reduced to SAT. For example, when each instance is described by just two tags, the reduction leads to 2-SAT, which is in **P** [10]. Likewise, if the treewidth of the resulting SAT formula is a constant, the problem can be solved efficiently; this follows from known results for SAT [3]. The intractability results for the *feasibility* of satisfying the `together` and `apart` constraints are different from those for constrained clustering [1]. In particular, finding a feasible description just for *one* cluster under `apart` constraints is intractable (see supplementary material) where as clustering under `apart` constraints is only intractable for any fixed $k \geq 3$ [7].

The cluster description problem allows taking an existing clustering obtained from one data set and explaining it with another. This is useful in a variety of situations. The data used to perform the clustering (e.g. a graph) may not be useful for explanation, the clustering may be historical (e.g. electoral maps) and need a modern explanation or problems where clusters are found on public data and explained with private data. We formulated the feasibility problem for finding cluster descriptors and established its intractability. We then explored an ILP formulation to incorporate this formulation and found there were some limitations: (i) many problem instances are infeasible and (ii) some explanations were inconsistent and incomplete. We addressed these two concerns by adding in a *cover-or-forget* ability so that some instances can be ignored if they are too difficult to cover and compositions constraints not unlike the `must-link` and `cannot-link` constraints used in clustering. Our ILP formulations scale to 10,000's of instances but not beyond. To address even larger problems, we created an $\alpha - \beta$ relaxation of the problem which is solvable in polynomial time when $\alpha$ and $\beta$ are fixed. It requires each cluster to be described by at most $\alpha$ tags and each pair of descriptions can have at most $\beta$ tags in common (overlap). Our experimental results in Twitter data show promising results and the usefulness of the cluster description problem. We can now use the Louvain method (applied to the `follower` graph) to find clusters and explain them using hashtags.

**Acknowledgments.** Ian Davidson was an Institute of Advanced Studies Fellow at the Collegium de Lyon at the time of writing and was also supported by Deep Graph Models of Functional Networks Grant ONR-N000141812485 and Functional Network Discovery NSF Grant IIS-1422218. S. S. Ravi was supported in part by NSF DIBBS Grant ACI-1443054, NSF BIG DATA Grant IIS-1633028 and NSF EAGER Grant CMMI-1745207. The twitter data was provided by the ERIC lab at the University of Lyon 2 and was prepared by one of the authors (Antoine Gourru). Thanks to Yue Wu (UC Davis) for writing the MATLAB code available at `www.cs.ucdavis.edu/~davidson/description-clustering/nips18_code`.

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
