[Supplementary Material]

# Supplementary Material

Paper title:    The Cluster Description Problem – Complexity Results,
                Formulations and Approximations

# 1    A Complexity Result for the DTDF Problem

**Theorem 1.1** *The* DTDF *problem is* **NP**-*complete even when the number of clusters is 2 and the tag set of each item is of size at most 3.*

**Proof:**    Membership in **NP** is obvious. We prove **NP**-hardness through a reduction from 3SAT which is known to be **NP**-complete [3]. Let $x_1$, $x_2$, ..., $x_n$ denote the $n$ variables and $Y_1$, $Y_2$, ..., $Y_m$ denote the $m$ clauses of the 3SAT instance. The reduction to the DTDF problem is as follows.

(a) For each variable $x_i$, we create two tags, denoted by $a_i$ and $b_i$. (Tags $a_i$ and $b_i$ correspond to the positive and negative literals of $x_i$). So, the tag set $T = \{a_1, a_2, \ldots, a_n, b_1, b_2, \ldots, b_n\}$, and $|T| = 2n$.

(b) For each variable $x_i$, we create an item $s_i$ with tag set $t_i = \{a_i, b_i\}$, $1 \le i \le n$. (Thus, $|t_i| = 2$, $1 \le i \le n$.) Items $s_1$, $s_2$, ..., $s_n$ constitute Cluster $C_1$.

(c) For each clause $Y_j$, we create an item $s_{n+j}$, $1 \le j \le m$. Suppose $Y_j$ contain literals $x_{j_1}$, $x_{j_2}$ and $x_{j_3}$. For each literal $x_{j_\ell}$ in $Y_j$, if $x_{j_\ell}$ corresponds to positive literal $x_i$, then $t_{n+j}$ contains $a_i$ and if $x_{j_\ell}$ corresponds to the negative literal $\overline{x_i}$, then $t_{n+j}$ contains $b_i$. (Thus, $|t_{n+j}| = 3$, $1 \le j \le m$.) Items $s_{n+1}$, $s_{n+2}$, ..., $s_{n+m}$ constitute Cluster $C_2$.

(d) The set of items $S = \{s_1, s_2, \ldots, s_{n+m}\}$.

Clearly, the above construction can be done in polynomial time. It can also be seen that the tag set of each item produced by the above construction is of size at most three.

Suppose there is a solution to the 3SAT instance. We construct tag sets $T_1$ and $T_2$ for clusters $C_1$ and $C_2$ as follows. For $1 \le i \le n$, if the given satisfying assignment sets variable $x_i$ to `True`, we add $a_i$ to $T_2$ and $b_i$ to $T_1$; if the given satisfying assignment sets variable $x_i$ to `False`, we add $b_i$ to $T_2$ and $a_i$ to $T_1$. It is easy to see that $T_1$ and $T_2$ are disjoint. Since the truth assignment satisfies all the clauses, $T_2$ has at least one item from each tag set $t_{n+j}$, $1 \le j \le m$. So, $T_1$ and $T_2$ constitute a solution to the DTDF problem.

Now suppose that there is a solution to the DTDF problem. We have the following claim.

**Claim 1:**    For each $i$, $1 \le i \le n$, $T_2$ contains at most one of $a_i$ and $b_i$.

**Proof of Claim 1:**    The proof is by contradiction. Suppose for some $i$, $1 \le i \le n$, $T_2$ contains both $a_i$ and $b_i$. Note that $C_1$ contains the item $s_i$ whose tag set is $\{a_i, b_i\}$. Thus, $T_1$ must contain at least one of $a_i$ and $b_i$. Now, since $T_1$ contains both $a_i$ and $b_i$, we conclude that $T_1$ and $T_2$ are not disjoint. This contradicts the assumption that we have a valid solution to the DTDF problem, and Claim 1 follows.

Given a solution to DTDF, we construct a solution to SAT as follows. Consider each variable $x_i$, $1 \leq i \leq n$. If tag $a_i \in T_2$, set $x_i$ to `True`. If $b_i \in T_2$ or neither $a_i$ nor $b_i$ appears in $T_2$, set $x_i$ to `False`. We claim that this is a valid satisfying assignment. First, using Claim 1, it is seen that each variable is set to either `True` for `False`. Consider any clause $C_j$. $T_2$ contains at least one of the tags from $t_{n+j}$, the tag set of item $s_{n+j}$ corresponding to $C_j$. Thus, the chosen assignment sets at least one of the literals in $C_j$ to `True`; that is, the clause is satisfied. This completes the proof of Theorem 1.1. ∎

## 2 A Complexity Result for the $(\alpha, \beta)$-Cons-Desc Problem

We showed in the main paper that when $\alpha$ (and hence $\beta$) and $k$ (the number of clusters) are *fixed*, the $(\alpha, \beta)$-Cons-Desc problem can be solved efficiently. We now show that when the number of clusters $k$ is *not fixed*, the $(\alpha, \beta)$-Cons-Desc problem remains **NP**-complete even when $\alpha$ is fixed.

**Theorem 2.1** *When the number of clusters $k$ is not fixed, The $(\alpha, \beta)$-Cons-Desc problem is* **NP**-*complete even when $\alpha = 4$ and $\beta = 1$.*

**Proof:** Membership in **NP** is obvious. We prove **NP**-hardness through a reduction from a restricted version of 3SAT in which each variable occurs either two or three times (considering both positive and negative literals of that variable) in the set of clauses. This restricted version of 3SAT, which will be denoted by R3SAT, is also known to be **NP**-complete [4].

Let $x_1$, $x_2$, ..., $x_n$ denote the $n$ variables and $Y_1$, $Y_2$, ..., $Y_m$ denote the $m$ clauses of the R3SAT instance. The reduction to the $(\alpha, \beta)$-Cons-Desc problem, where $\alpha = 4$ and $\beta = 1$, is as follows.

1. We first describe how the data items of the $(\alpha, \beta)$-Cons-Desc instance are produced.

   (a) For each variable $x_i$ ($1 \leq i \leq n$), we create a data item $w_i$. Let $W = \{w_1, w_2, \ldots, w_n\}$.

   (b) For each clause $Y_j$, ($1 \leq j \leq m$), we create a data item $p_j$. Let $P = \{p_1, p_2, \ldots, p_m\}$.

   (c) Recall that in R3SAT, each variable occurs positively or negatively in either two or three clauses. Consider each variable $x_i$. If $x_i$ occurs three times in R3SAT, we create six data items denoted by $d_i^1$, $e_i^1$, $d_i^2$, $e_i^2$, $d_i^3$ and $e_i^3$. If $x_i$ occurs two times in R3SAT, we create only the first five of these data items (i.e., we don't create $e_i^3$). For each $i$, $1 \leq i \leq n$, we will refer to these six or five data items as the *special* data items corresponding to variable $x_i$. Let $D$ denote the set of all data items created in this step. (Thus, each data item in $D$ is a special data item corresponding to some variable of R3SAT.)

   (d) The set $S$ of data items for the $(\alpha, \beta)$-Cons-Desc problem is given by $S = W \cup P \cup D$.

2. Next, we describe the construction of the set of tags for each data item created above.

   (a) The tag set $\tau(w_i)$ for each data item $w_i \in W$ has two tags, denoted by $a_i$ and $b_i$. (Tags $a_i$ and $b_i$ correspond to the positive and negative literals of $x_i$).

   (b) Consider each data item $p_j \in P$ corresponding to clause $Y_j$. The tag set $\tau(p_j)$ for the data item $p_j$ has three tags chosen as follows. Suppose $Y_j$ contain literals $x_{j_1}$, $x_{j_2}$ and $x_{j_3}$. For each literal $x_{j_\ell}$ in $Y_j$, if $x_{j_\ell}$ corresponds to positive literal $x_i$, then $a_i$ is added to $\tau(p_j)$ and if $x_{j_\ell}$ corresponds to the negative literal $\overline{x_i}$, then $b_i$ is added to $\tau(p_j)$.

(c) For each variable $x_i$, $D$ contains five or six special data items. First consider the case where $x_i$ has six special data items, namely $d_i^1$, $e_i^1$, $d_i^2$, $e_i^2$, $d_i^3$ and $e_i^3$. For each $\ell$, $1 \le \ell \le 3$, the tag sets $\tau(d_i^\ell)$ and $\tau(e_i^\ell)$ contain just one tag, denoted by $t_i^\ell$. If $x_i$ has five special data items, we do the same construction except we don't produce a tag set for $e_i^3$ (since that data item doesn't exist). We will refer to the set $\{d_i^1, d_i^2, d_i^3\}$ as the **primary special item set** corresponding to $x_i$ and the set $\{e_i^1, e_i^2, e_i^3\}$ (or $\{e_i^1, e_i^2\}$ when $x_i$ appears only in two clauses) as the **secondary special item set** corresponding to $x_i$.

3. We now describe the construction of the clusters.

   (a) For each variable $x_i$, we have a cluster $A_i$ consisting $w_i$ and all the three data items $d_i^1$, $d_i^2$ and $d_i^3$ from the primary special item set corresponding to $x_i$, $1 \le i \le n$. (Thus, $|A_i| = 4$, $1 \le i \le n$.)

   (b) For each clause $Y_j$, we have a cluster $B_i$ containing the following data items. First, cluster $B_j$ includes the data item $p_j$. Suppose $Y_j$ contain (positive or negative) literals of variables $x_{i_1}$, $x_{i_2}$ and $x_{i_3}$. Then, $B_j$ also contains one arbitrary data item each from the secondary special item sets corresponding to the variables $x_{i_1}$, $x_{i_2}$ and $x_{i_3}$. Thus, $|B_j| = 4$, $1 \le j \le m$. It should be noted that each data item in the secondary special item set of each variable $x_i$ can only be used in one cluster $B_j$. (This is because the clusters must form a partition of the data set $S$.)

   (c) The set $C$ of $n + m$ clusters produced by the construction is given by $C = \{A_1, \ldots, A_n, B_1, \ldots, B_m\}$.

   Note that For each variable $x_i$, the construction five or six special data items in $D$. Three of these data items (i.e., the primary special data items corresponding to $x_i$) are in cluster $A_i$. each of the remaining special data items (i.e., those in the secondary special data item set of $x_i$) appears in one cluster corresponding to each clause in which variable $x_i$ occurs.

This completes the construction. It can be verified that the construction can be carried out in polynomial time. We observe that each cluster has exactly four data items.

We now show that there is a solution to the $(\alpha, \beta)$-CONS-DESC problem with $\alpha = 4$ and $\beta = 1$ iff there is a solution to the R3SAT problem.

Suppose there is a solution to the R3SAT instance. We construct a tag set for each cluster as follows.

(a) Consider each cluster $A_i$ $(1 \le i \le n)$. For the data item $w_i \in A_i$, tag set is $\{a_i, b_i\}$. If variable $x_i$ is assigned the value True, we choose $b_i$ in the descriptor for $A_i$; otherwise, we choose $b_i$. The other three data items in $A_i$ are from $D$, and each has only one unique tag. We add those three tags to the descriptor set for $A_i$.

(b) Consider each cluster $B_j$ $(1 \le j \le m)$. For the data item $p_j \in B_j$, tag set is created from the literals in clause $Y_j$. Since the satisfying assignment makes at least one of the literals in $Y_j$ to be True, we pick the tag corresponding to an arbitrary literal that is set to Trueby the assignment. The other three data items in $B_j$ are from $D$, and each has only one unique tag. We add those three tags to the descriptor set for $B_j$.

It is not difficult to verify that each descriptor set has exactly four tags and that any two descriptors have at most one tag in common. In other words, we have a solution to the $(\alpha, \ \beta)$-CONS-DESC problem with $\alpha = 4$ and $\beta = 1$.

Now, suppose there is a solution to the $(\alpha, \ \beta)$-CONS-DESC problem. We show how to construct a satisfying assignment to the R3SAT instance. Consider the clause $A_i$ corresponding to the Boolean variable $x_i$ $(1 \leq i \leq n)$. $A_i$ contains data item $w_i$ with tag set $\{a_i, b_i\}$ and three other data items from $D$, and each of those three data items has a unique tag. So, the descriptor for $A_i$ must have those three tags. Since the descriptor can only at most four tags, it can include exactly one of $a_i$ and $b_i$. If the descriptor for $A_i$ includes $a_i$, we set variable $x_i$ to False; otherwise, we set $x_i$ to True. We now argue that this is a satisfying assignment for each of the clauses. Consider any clause $Y_j$ and the corresponding cluster $B_j$. The descriptor for $B_j$ must include the three tags corresponding to the data items from the set $D$ in $B_j$ since each of those three items has a unique tag. Since the descriptor for $B_j$ is of size 4, it can only include one of the tags of the data item $p_j \in B_j$. Suppose the chosen tag is $a_r$ corresponding to the literal $x_r$. (The proof is similar if the chosen tag is $b_r$ corresponding to the literal $\overline{x_r}$.) Thus $x_r$ occurs in clause $Y_j$. We prove by contradiction that the chosen assignment sets $x_r$ to True. Suppose $x_r$ is set to False. Consider the descriptors for $B_j$ and $A_r$ (the clause corresponding to $x_r$). Since $x_r$ is set to False, the descriptor for $A_r$ must contain $a_r$. Note that the variable $x_r$ appears (as a positive literal) in $Y_j$. Thus, there is a pair of data items $d_r^h$ and $e_r^h$ such that $d_r^h \in A_r$, $e_r^h \in B_j$ and this pair of data items has the same unique tag, namely $t_r^h$. So, the descriptor sets of $B_j$ and $A_r$ have one common tag, namely $t_r^h$. Further, since $x_r$ is set to False, the two descriptor sets also have the tag $a_r$ in common. In other words, the overlap between the descriptors of $B_j$ and $A_r$ is at least two, contradicting the assumption that $\beta = 1$. Hence, the truth assignment must set $x_r$ to True, and clause $Y_j$ is satisfied. In other words, we have a solution to the R3SAT instance, and this completes the proofs of Theorem 2.1. ∎

## 3   Finding Descriptors Under Apart (or Cannot-Link) Constraints

We use CL-FEASIBILITY to denote the feasibility problem under Apart (also called cannot-link or CL) constraints. We show that CL-FEASIBILITY is computationally intractable even for a single cluster.

**Theorem 3.1** *Given a single cluster $L$ and a set $A$ of CL constraints, the problem of determining whether there is a descriptor for $L$ that satisfies all the constraints in $A$ is* **NP**-*complete.*

**Proof:**  It is easy to see that CL-FEASIBILITY is in **NP**. Our proof of **NP**-hardness uses a reduction from 3SAT. This reduction is similar to the one used to prove Theorem 1.1.

Let $x_1$, $x_2$, ..., $x_n$ denote the $n$ variables and $Y_1$, $Y_2$, ..., $Y_m$ denote the $m$ clauses of the 3SAT instance. The reduction to the DTDF problem is as follows.

(a) For each variable $x_i$, we create two tags, denoted by $a_i$ and $b_i$. ($a_i$ and $b_i$ correspond to the positive and negative literals of $x_i$). So, the tag set $T = \{a_1, a_2, \ldots, a_n, b_1, b_2, \ldots, b_n\}$, and $|T| = 2n$.

(b) For each clause $Y_j$, we create an item $s_j$, $1 \leq j \leq m$. Thus, the set of items $S = \{s_1, s_2, \ldots, s_m\}$. The tag set $t_j$ for $s_j$ is chosen as follows. Suppose $Y_j$ contain literals $x_{j_1}$, $x_{j_2}$ and $x_{j_3}$. For

each literal $x_{j_\ell}$ in $Y_j$, if $x_{j_\ell}$ corresponds to positive literal $x_i$, then $t_j$ contains $a_i$ and if $x_{j_\ell}$ corresponds to the negative $\overline{x_i}$, then $t_j$ contains $b_i$. (Thus, $|t_j| = 3$, $1 \leq j \leq m$.) The set $S = s_1, s_2, \ldots, s_m$ constitutes the only Cluster $L$.

(c) The constraint set $A$ has $n$ CL-constraints given by $\mathrm{CL}(a_i, b_i)$, $1 \leq i \leq n$.

It can be seen that the above construction produces just one cluster. Further, the cardinality of the tag set for each item is exactly three.

Suppose there is a solution to the 3SAT instance. we construct a tag set $Q$ for the cluster $L$ as follows. For $1 \leq i \leq n$, if the given satisfying assignment sets variable $x_i$ to True, we add $a_i$ to $Q$; otherwise, we add $b_i$ to $Q$. Note that for each $i$, $Q$ contains exactly one of $a_i$ and $b_i$, $1 \leq i \leq n$. Thus, all the CL constraints in $A$ are satisfied. Since the truth assignment satisfies all the clauses, $Q$ has at least one item from each tag set $t_j$, $1 \leq j \leq m$. So, $Q$ constitutes a solution to the CL-FEASIBILITY problem.

Now suppose that there is a solution to the CL-FEASIBILITY problem. Let $Q$ denote the chosen descriptor for the cluster $L$. For each $i$ ($1 \leq i \leq n$), if $a_i \in Q$, we set $x_i$ to True and if $b_i \in Q$, we set $x_i$ to False. Further, if neither $a_i$ nor $b_i$ appears in $Q$, we set $x_i$ to False. We first note that this assigns a truth value to each variable $x_i$. Further, since the CL constraints ensure that for each $i$, $Q$ does not contain both $a_i$ and $b_i$, each variable is assigned a unique truth value. We now claim that this assignment satisfies all the clauses. To see this, consider any clause $Y_j$. Note that $Q$ contains at least one of the tags from $t_j$, the tag set of item $s_j$ corresponding to $Y_j$. Thus, the chosen assignment sets at least one of the literals in $Y_j$ to True; that is, the clause is satisfied. This completes the proof of Theorem 3.1. ∎

We note that the cluster description problem with CL constraints differs significantly from the feasibility problem for finding clusters with CL constraints. In particular, the feasibility problem for finding clusters satisfying a given set of CL constraints is efficiently solvable for two clusters while it is **NP**-complete for three or more clusters [2]. For the cluster description problem, computational intractability sets in even for the simplest case, namely describing a single cluster.