[Reviews · NeurIPS 2018]

Reviewer 1



This paper examines the problem of cluster description, in which the clustering algorithm has access to one view of the data, and we would like to explain its output using another view of the data. The problem is interesting and well-motivated. The authors begin by formulating the problem in terms of clusters and "tags". We wish to describe each disjoint cluster by a disjoint set of tags such that each member of the cluster has at least one of the tags. Perhaps unsurprisingly given its similar appearance to the set cover problem, this is shown by the authors to be NP-complete (though some special subclasses are solvable in polynomial time). The authors then introduce an extended formulation which requires that the size of the descriptor tag sets is minimized. While still intractable, the formulation lends itself to an ILP. While this ILP often leads to poor explanations of the clustering, the authors introduce the ability to omit data instances and add constraints on which tags can be used together. Finally, the authors consider the natural relaxation of the cluster description problem in which the tag sets for different clusters need not be disjoint -- but must have a small intersection. It is shown that this relaxation permits a naive polynomial-time algorithm provided that the number of clusters is fixed. Overall, the paper is clearly written and its results appear sound. Unfortunately, I am not well-equipped to judge the impact that it is likely to have; the NP-completeness of the first formulation perhaps isn't so surprising, and the experiments with the Twitter election data seem to require a large number of constraints to produce satisfactory output. I'll therefore defer to other reviewers who might better be able to ascertain impact.

Reviewer 2



The main contributions of the paper are: - Formulating a cluster description problem as a combinatorial optimisation problem to find descriptions for clusters that are descriptive and non-overlapping - Showing NP-completeness of the problem, even if there are only two clusters - Providing ILP encodings modelling (also extended) variants of the problem - Providing a tractable subclass of the problem, and together with a polynomial time algorithm for this subclass Pros: - A novel problem that is relevant by providing explanations for previously computed clusterings - Well-written paper with a clear focus and clear contributions Cons: - In lines 102–103, it is stated that some version of DTDF can be reduced to SAT. Should not this be 2-SAT? Since DTDF is shown to be NP-complete, all versions of DTDF can be reduced to SAT, right? The NP-completeness should hold in general (if not, it should not be possible to devise and ILP encoding for the problem). - The experimental section is missing some detail (see minor comments below) Quality: The paper is technically sound and the proofs for complexity parts are provided in the supplementary material. While I did not check all the details in the supplementary, the constructions for reductions seem sound. The cluster description problem is solved in general using a ILP formulation and also for a tractable subclass. The experimental evaluation supports the theoretical results. Clarity: The paper is well written and I found it a pleasure to read. Being a reader with a strong background in combinatorial methods I found the level of details sufficient and discussion very concise, and I think that non-experts on combinatorial optimisation should be able to follow the paper. Originality: The problem definition is novel to my knowledge. The work is placed well in context with other (not too similar) works. Significance: The results and the new problem formulation have potential for significance. It is relevant for us to find new methods to understand and explain the solutions provided by machine learning algorithms. Minor: -line 61: descriptive -> description - the Twitter example with hashtags related to Trump-Clinton election could use some explanation about the meanings of hashtags etc. While currently these should be clear to many if not all, in couple of years it might not be as clear why #MakeAmericaGreatAgain should not go together with #ImWithHer. - It was not really clear to me, whether the IPL encoding or Algorithm 1 was used in the experimental section. - Regarding Figure 3: how many tags were there in the left table? How many nodes were there in the right table? What is the meaning on *? Is the number of nodes forgotten correct for 25 tags and k=60? AFTER AUTHOR RESPONSE: The response answered questions raised in the review in a satisfactory manner.

Reviewer 3



The paper introduces a particular type of post-clustering problem, where one needs to tag each cluster under instance-level constraints (in particular, co-clustering is not possible in their framework). While the paper seems to be technically rigorous and interesting, my main concern is how general (re-usable) this problem formulation is (the paper is mainly motivated by the twitter tagging problem; in the introduction it also mentions potential usage for satellite imagery and medical data, but without a detailed explanation or a reference to either the dataset or the problem, it is hard to judge how realistic these problems are.